# A Century of Progress on Wilson Disease and the Enduring Challenges of Genetics, Diagnosis, and Treatment

**DOI:** 10.3390/biomedicines11020420

**Published:** 2023-02-01

**Authors:** Louis C. Penning, Marina Berenguer, Anna Czlonkowska, Kay L. Double, Petr Dusek, Carmen Espinós, Svetlana Lutsenko, Valentina Medici, Wiebke Papenthin, Wolfgang Stremmel, Jose Willemse, Ralf Weiskirchen

**Affiliations:** 1Department of Clinical Sciences of Companion Animals, Faculty of Veterinary Medicine, Utrecht University, 3584 CM Utrecht, The Netherlands; 2Digestive Medicine Department, Ciberehd & IISLaFe, Hospital U. i P. La Fe, University of Valencia, 46010 Valenci, Spain; 3Second Department of Neurology, Institute of Psychiatry and Neurology, 02-957 Warsaw, Poland; 4Brain and Mind Centre and School of Medical Sciences (Neuroscience), The University of Sydney, Sydney, NSW 2006, Australia; 5Department of Radiology, Charles University and General University Hospital, 128 08 Prague, Czech Republic; 6Department of Neurology and Centre of Clinical Neuroscience, First Faculty of Medicine, Charles University and General University Hospital, 128 08 Prague, Czech Republic; 7Rare Neurodegenerative Diseases Lab, Centro de Investigacion Principe Felipe, 46012 Valencia, Spain; 8Department of Medicine, Johns Hopkins University School of Medicine, Baltimore, MD 1800, USA; 9Department of Physiology, Johns Hopkins University School of Medicine, Baltimore, MD 1800, USA; 10Department of Internal Medicine, Division of Gastroenterology and Hepatology, University of California Davis, Sacramento, CA 59817, USA; 11German Society for Wilson disease Patients (Morbus Wilson e.V.), Zehlendorfer Damm 119, D-14532 Kleinnachnow, Germany; 12Private Practice for Internal Medicine, Beethovenstraße 2, D-76530 Baden-Baden, Germany; 13Dutch Society for Liver Disease Patients (Nederlandse Leverpatienten Vereniging), 3828 NS Hoogland, The Netherlands; 14Institute of Molecular Pathobiochemistry, Experimental Gene Therapy and Clinical Chemistry (IFMPEGKC), RWTH Aachen University Hospital Aachen, D-52074 Aachen, Germany

**Keywords:** Wilson disease, copper accumulation, hepatic disfunction, neurological disfunction, psychiatric disorders, patients’ involvement

## Abstract

Wilson disease (WD) is a rare, inherited metabolic disorder manifested with varying clinical presentations including hepatic, neurological, psychiatric, and ophthalmological features, often in combination. Causative mutations in the *ATP7B* gene result in copper accumulation in hepatocytes and/or neurons, but clinical diagnosis remains challenging. Diagnosis is complicated by mild, non-specific presentations, mutations exerting no clear effect on protein function, and inconclusive laboratory tests, particularly regarding serum ceruloplasmin levels. As early diagnosis and effective treatment are crucial to prevent progressive damage, we report here on the establishment of a global collaboration of researchers, clinicians, and patient advocacy groups to identify and address the outstanding challenges posed by WD.

## 1. Introduction

The first description of what is now known as the family of Wilson disease (WD) was published in a French-language scientific journal *La Revue Neurologique* in 1912, and around the same time, in the English-language journal *Brain* [1,2]. These reports described patients with both neurological and hepatic presentations. Slightly later, copper was proposed to be crucial in the pathogenesis of WD [3,4], and in the middle of the 20th century, copper chelators, such as D-penicillamine, were introduced in clinical practice (for an overview of historical landmarks in WD research, see [5,6]). An initial report of the gene responsible for WD on chromosome 13 (q21.1–14.3) was published in 1985, yet it took another eight years before several groups independently confirmed the *ATP7B* gene as the causative gene for WD [7,8,9,10,11]. The identification of this gene raised hopes for the development of a rational gene-based therapeutic strategy. The translation of this molecular advance into clinical benefits has proved to be more complicated than expected. Despite the availability of genetic screens, clinical diagnosis remains challenging. The WD genotype does not predict disease phenotype.

To achieve the next step in the fight against WD, an international collaborative, i.e., the International Wilson’s disease consortium, was established with the goal of reaching a consensus regarding the outstanding research questions that should be prioritized in the next decade(s) to positively impact the lives of people living with WD. Opinions from neurologists and hepatologists with a keen interest in WD were combined with those of basic research experts in the fields of copper transport, cellular/organ copper distribution, genetics of rare diseases, and animal models of disease. The major challenges identified by the network were discussed with WD patient advocacy groups to ensure that the identified goals aligned with the unmet clinical needs experienced by WD patients. The description of specific clinical cases illustrates the most important outstanding questions (Box 1, Box 2, Box 3, Box 4, Box 5, Box 6 and Box 7). In addition, a plea for improved copper measurements (Box 8) and a peculiar non-human copper toxicosis patient group is presented (Box 9), which has been very useful to discover modifier genes affecting the phenotypical presentation of hepatic copper accumulation. Last but not least, a patients’ perspective and the role of patient advocacy groups are illustrated (Box 10 and Box 11). The clinical cases and other boxes are presented as examples to highlight everyday problems encountered by (interactions between) patients, clinicians, and scientists. As such, these are not intended as detailed descriptions or solution-providing cases. This is not the first review or perspective on WD and will surely not be the last. Moreover, at present, 10 European experts are working to update the clinical guidelines for WD. However, to our knowledge, this is the first manuscript where clinicians, scientists, patients, and patient advocacy groups combine forces to pinpoint their opinions on the directions for WD investigations in the next decade. Hopefully, this innovative multi-angle initiative will stimulate and deepen collaborations to beat WD.

Box 1Wilson disease alone is not the full story.Wolfgang Stremmel and Ralf Weiskirchen  The importance of modifier genes is apparent from the clinical observation that the course of Wilson disease can vary markedly, despite an identical pattern of mutations. However, comorbid conditions may also impact disease manifestation.  A 9-year-old girl was diagnosed with hepatic Wilson disease in the absence of neurological symptoms and was treated with D-penicillamine. One year later, transaminases were normalized, and free serum copper and urinary copper were within normal ranges. Therapy was maintained but transaminases were observed to rise a year later. D-penicillamine was changed to trientine, and later to zinc, but neither influenced transaminase levels, which continued to rise. Right from the start of her illness before any therapy, elevated antinuclear antibodies had been noted without hyperimmunoglobulemia prior to the initiation of any treatment but had been discounted as a potential disease-contributing factor. Due to persistent liver inflammation, an explorative short-term course of prednisolone was trialed. Transaminases dropped dramatically from >500 to 84 U/Lwithin 2 months. This case highlights that a hyperimmune state without elevation of immunoglobulins concurrent with Wilson disease may predispose to liver inflammation.Outstanding questions:Does a hyperimmune state with isolated elevation of autoantibodies predispose to liver inflammation?Is such hyperimmunity more frequent in Wilson disease?Is manifestation of liver disease frequently a multifactorial process?

Box 2Diagnostic testing for Wilson disease: still a matter for debate?Valentina Medici  A 24-year-old woman presented with elevated levels of transaminases and bilirubin and increased blood clotting times. Ceruloplasmin levels were borderline at 20 mg/dL. Antinuclear antibodies (ANAs) and antibodies to smooth muscle actin (ASMAs) titers were suggestive of an autoimmune response (both 1:40). An initial liver biopsy showed interface hepatitis, portal and lobular inflammation, and hepatocyte necrosis. The patient was diagnosed with autoimmune hepatitis and started on azathioprine and steroids. However, after 2 months of immunosuppressive treatment at recommended doses, liver enzyme levels were only partially improved; therefore, other diagnoses were considered. Ceruloplasmin levels were again quantified and were 19 mg/dL, and 24 h urinary copper excretion was 350 µg. A repeat liver biopsy demonstrated liver copper levels to be 450 µg/g DWL. A diagnosis of Wilson disease was established, and the patient was started on copper chelator therapy with improvements in liver enzymes after 2 months at a full regimen consisting of 1500 mg/day.  This case demonstrates that Wilson disease can present with laboratory and histological features, as well as clinical presentations, similar to autoimmune hepatitis. These shared features can lead to diagnostic delays and thus a delayed anti-copper treatment response.Outstanding questions:Can we identify a gold-standard for diagnosis of Wilson disease?Are currently available tests adequate to promptly distinguish Wilson disease from other chronic liver diseases?

Box 3What are the best strategies for Wilson disease during aging?Valentina Medici  A 67-year-old man was diagnosed with Wilson disease at age 35. The diagnosis was based on mildly elevated liver enzymes, low ceruloplasmin levels (9 mg/dL) and increased 24-hour urinary copper excretion at 270 µg. A liver biopsy at baseline demonstrated mild portal inflammation. During initiation of treatment with the copper chelator D-penicillamine, liver enzymes normalized and remained within normal range such that a liver biopsy was not repeated. In later life, however, he gained weight, gradually reaching a BMI of 32, and developed type 2 diabetes, hypertension, and chronic kidney disease stage 2, which was attributed to his diabetes. As penicillamine was apparently tolerated and liver enzymes remained normal, his anti-copper treatments were not tailored to his changing comorbidities.Outstanding questions:Wilson disease affects pediatric and young adults, but the patient population is aging with associated comorbidities. What is the best anti-copper treatment approach in elderly patients?When should we consider a change in anti-copper treatment strategy?How do we monitor any treatment transition? When can we consider a treatment change to be successful?

## 2. Specific Problems Related to Rare Diseases

WD, a rare metabolic disease, is an autosomal recessive disorder associated with aberrant copper metabolism resulting in excessive copper accumulation, primarily in the brain and liver [6]. Defined (by European criteria) as a rare disease, WD has a prevalence of lower than 1 in 2000 (<50 per 100,000) individuals [12]. This definition differs slightly from the US criteria for a rare disease (RD), which is <86 per 100,000. Despite these differences, and accounting for the 6000–7000 unique RDs described to date, the total number of people affected by RDs is estimated to be well over 300 million [9]. This corresponds to approximately 5% of the world population [13], or 6–8% of the European population. Almost 75% of described RDs have a genetic background, and they typically present at young age [14].

RDs, such as WD, are associated with specific problems for the patient. Typically, patients visit numerous medical specialists after consulting a general practitioner before a final diagnosis is achieved. In addition, patients with RDs may suffer from a lack of social support, as the social environment often does not recognize their experience. Symptoms may be judged as induced by lifestyle choices or, even more of a concern, as self-inflicted [13]. It may take considerable time, for example, a decade, before an accurate diagnosis can be reached, by which time irreversible damage may have already occurred. Even when a diagnosis is finally reached, options for management or treatment may be limited. A lack of large, well-characterized patient cohorts and small research funding often results in a scarcity of research on specific RDs. To address these challenges for rare hepatological diseases, the European Commission established a European Reference Network for rare liver diseases [15], while the International Rare Diseases Research Consortium (IRDiRC) was created to increase scientific knowledge of all RDs. The Orphanet database provides publicly available epidemiological data on RDs [13]. Together, these investments by the European Commission highlight the priority assigned by the European Union to improve the knowledge of RDs and care of individuals experiencing these disorders.

## 3. Intracellular Copper Transport

Hepatic or neuronal copper accumulation is the pathogenetic basis of WD. Copper acts as a ‘Jekyll and Hyde’ transition element. On the one hand, this trace element is indispensable for the function of numerous key proteins, such as lysyl oxidases (cartilage synthesis), aminoxidases (oxidation of primary amines to aldehydes), and the superoxide dismutases (antioxidant defense). On the other hand, it is involved in the creation of reactive oxygen species via Fenton and Haber–Weiss chemistry. As a result, it is important to keep the intrahepatic Cu^+^ (cuprous) concentration within the normal range of 15–55 µg/g dry weight liver tissue (DWL). A concentration above the established cutoff, which is 250 µg/g DWL, fulfills a diagnostic criterion for WD. Following intake of copper via the diet (recommended daily intake of copper is age-dependent, e.g., for adults, the recommended intake is 900 µg/day of cupric copper (Cu^2+^)), reduced Cu^+^ is taken up in the gastrointestinal tract and excreted via the action of ATPase7A in enterocytes. It is subsequently transported through the bloodstream, reaching the liver via the portal vein, where copper transporter 1 (CTR1) mediates uptake in hepatocytes. Once inside hepatocytes, Cu^+^ can be sequestered by copper chaperone for the superoxide dismutases (CCS) or can bind to metallothionein (MT) or Parkinson disease protein 7 (PARK7), all of which represent stable binding and thus detoxification of Cu^+^.

Other routes for intrahepatic copper include cytochrome c oxidase copper chaperone (COX17)-mediated transport to either synthesis of cytochrome c oxidase 1/2 (SCO1/2) or cytochrome c oxidase 11 (COX11), which mediates the involvement of copper in mitochondrial oxidative phosphorylation. Lastly, antioxidant protein 1 (ATOX1) distributes Cu^+^ to the endoplasmic reticulum-bound ATPases, where, in situations of copper excess, biliary copper excretion is increased via transport by ATP7B translocated with the aid of copper metabolism Murr1 domain 1 (COMMD1). ATP7B also mediates the release of ceruloplasmin (CP)-bound copper into the bloodstream. The CP-copper complex mediates iron oxidation (Fe^2+^ to Fe^3+^) to facilitate Fe^3+^ binding to ferritin (Figure 1).

The vast majority (±40%) of dietary copper is transported to the bone, 20% to muscles, and around 10% to heart and brain tissue. The reader is referred to numerous reviews on the topic of intracellular copper homeostasis [17,18,19,20,21,22,23,24,25].

The central role of the ATP7B protein in the excretion of excessive amounts of intracellular copper is evidenced by the large number of mutations in the *ATP7B* gene and their relationship to development of WD. As previously described, WD is an autosomal recessive disorder resulting from mutations in the *ATP7B* gene located on chromosome 13, which consists of 21 exons and 20 introns. In humans, the ATP7B protein is 1465 amino acids long. The *N*-terminal copper binding domain with six copper binding sites resides in the cytosol. The phosphatase domain is located between the fourth and fifth transmembrane regions, and the ATP-binding domain is located between transmembrane domains 6–7 also faces the cytosol. The *C*-terminal does not have a described function (Figure 2).

Intracellular ATP7B transports copper bound to ATOX1 into the trans-Golgi network (TGN). Once released in the TGN, copper binds to apoceruloplasmin and is subsequently released at the basolateral side of the hepatocyte into the blood stream. Alternatively, ATP7B directs the exocytosis of copper-laden vesicles into the bile canaliculi. The lack of a functional ATP7B protein results in the enhanced binding of intracellular copper to MT proteins. Once these proteins are overloaded, copper accumulates in lysosomes that eventually rupture, leading to intracellular damage, for example, in the mitochondria [27,28]. An increasing number of mutations in the *ATP7B* gene have been identified; currently, around 1000 mutations are known, mostly in coding, intronic, and regulatory sequences, all of which affect the function and/or expression of *ATP7B* [29,30,31]. For the most up-to-date list of *ATP7B* gene mutations, see [32] or HGMD-professional, which lists (5 May 2022) 1112 deleterious mutations [33]. The vast majority (>65%) of all mutations are missense/nonsense mutations or small deletions (±15%), while the most frequent variant is p.H1069Q, which affects the ATP-binding domain at the cytosolic side of the 8-transmembrane protein ATP7B. This variant has a shorter half-life compared to the wild-type ATP7B [31]. Another frequent mutation is p.R778L, which is especially prevalent in Asian populations. This mutation occurs in the first of the eight transmembrane regions of ATP7B, altering the adjacent copper binding *N*-terminal domain of ATP7B, with consequent aberrant transmembrane copper transport.

## 4. Genetics and Epidemiology

Based on health registrations, the global incidence of WD is reported to be 1 in every 30,000 births. This prevalence is likely to be an underestimation; however, early diagnosis is difficult and carrier frequency is estimated to be around 1:5000 based on larger population-wide genetic screens [6,34,35,36], indicating that the true incidence is probably higher.

Higher prevalence rates are reported in some regional populations such as Sardinia (Italy) and on the Canary Islands (Spain) [37,38,39,40,41,42,43]. Modern techniques, such as percentage of penetrance, state-of-art sequencing (either whole exome sequencing (WES), or whole-genome sequencing (WGS)) may improve diagnosis of WD, as well as identify disorders which mimic WD, or novel variants, including novel modifier genes and epigenetic factors [44,45,46]. This raises the question of the need for population and/or newborn screenings for *ATP7B* variants.

The contribution of specific *ATP7B* mutations on the clinical presentation of neurological and/or psychiatric symptoms in WD is a matter of discussion. Other factors, including lifestyle and environmental factors, as well as inheritable factors, such as epigenetics and modifier genes, are also thought to contribute to the variable clinical presentation of WD. Several established modifier genes for WD have been shown to partially explain some patient-specific aspects of the clinical presentation [6,47,48,49]. Further, disturbed lipid homeostasis observed in a series of WD patients has been attributed to the genes coding for PNPLA3 or APOE, although this has not been confirmed in independent cohorts [50,51,52].

Two proteins known for their involvement in copper homeostasis have been suggested as modifier gene products, namely ATOX1 and COMMD1. Mutations in ATOX1 could affect its interaction with ATP7B and thus modify copper transport [53,54,55,56]. Mutations in the *COMMD1* gene, causative for inherited copper toxicosis in the dog breed Bedlington terrier [57], do not appear to be modifier genes in WD [54,58,59,60] and do not have a role for the anti-apoptotic protein X-chromosome-associated apoptosis inhibitor (XIAP), which interacts with COMMD1, has been identified in the age-related onset of WD [61].

The transmembrane copper transporter CTR1 also transports iron; therefore, two other proteins involved in iron transport were investigated as potential modifier genes in WD: homeostatic iron regulator (HFE) and divalent metal transporter 1 (DMT1). No clear correlation between variable phenotypes and mutations in either one of these genes was observed, although a specific *HFE* mutation appeared to be correlated with a poor response to copper chelation therapy in a small WD cohort in Sardinia [62,63,64,65,66]. Two polymorphisms in the methylenetetrahydrofolate reductase (MTHFR) gene are reported to be linked to the age of onset, although this has not been confirmed in an independent cohort [67,68].

Investigations of candidate modifier genes for WD in patients have been disappointing to date, but findings in other species suggest genetic interactions may be critical. An example of a novel DNA-sequencing based approach to identify modifier genes was presented in a WGS study on Labrador retrievers, in which copper toxicosis occurs at high frequency [69,70]. Copper toxicosis in Labrador Retrievers results from mutations in the *ATP7B* gene, but this etiology now appears to be modified by variations in the *ATP7A* gene [71].

## 5. Hepatic and/or Neurological Presentation

The two most prominent clinical signs result from copper accumulation in hepatocytes and copper deposition in the basal ganglia as well as other brain regions. Elevation of liver enzymes in the absence of concurrent liver pathology is a clear sign of liver damage. Interestingly, in addition to acute liver failure and cirrhosis, lipid accumulation (steatosis) in hepatocytes and steatosis also frequently occur. Elevated cerebral copper may result in neurological manifestations, including movement disorders, dysarthria, dysphagia, sialorrhea, cognitive dysfunction, and psychiatric symptoms. WD patients can express Kayser-Fleischer’s rings in the cornea as a result of copper accumulation, but not all WD patients do. Hemolytic anemia can also result from the sudden release of iron and copper from damaged erythrocytes. Urinary copper excretion is also often elevated, accompanied by tubular dysfunction and aminoaciduria. Because copper is essential for the activity of lysyl oxidases involved in cartilage function, patients may also present with osteoarthritis or osteoporosis. The age of onset and sex also appear to influence phenotype.

Moreover, pregnant women need close monitoring and multidisciplinary management because pregnancy induced complication in WD such as prematurity, hypertension, preeclampsia, and gestational diabetes mellitus are very frequent [72]. Therefore, respective patients need adequate medical treatment and close monitoring before and during pregnancy to achieve the best health outcomes for the mother and newborn affected by WD.

In addition, a purely hepatic presentation is primarily observed in pediatric patients, whereas adult-onset patients more commonly express a mixed hepatic/neurological disease. Further, hepatic presentation is more common in females, with neurological impairments more common in male patients [73].

The Leipzig scale was developed to assist the diagnosis of WD in 2012 [74]. This diagnostic algorithm is based on a range of clinical signs indicative of WD, including the presence of Kayser-Fleischer’s rings, the presence and severity of neurological symptoms, reduced levels of ceruloplasmin, Coombs-negative hemolytic anemia, liver copper levels (absolute levels, or indicated by Rhodamine-staining), elevated urinary copper, and *ATP7B* mutations.

Currently, the presence of Kayser-Fleischer’s rings is the most reliable sign of WD, especially when combined with reduced serum ceruloplasmin levels. Inexperience can lead to false diagnosis here. However, the absence of Kayser-Fleischer’s rings or near-normal serum ceruloplasmin levels does not exclude WD. Whereas copper cannot be directly quantified in the living brain, MRI may depict the presence of pathology in the WD brain. Such pathology, typically T2 hyperintense lesions in the deep grey matter and brainstem, is assumed to be associated with copper deposition in the brain and is thus included in the Leipzig scale. Interestingly, pathology in the WD brain observed by MRI may be present in the absence of overt neurological symptoms. On the other hand, patients may present with cognitive or psychiatric symptoms in the absence of obvious morphological changes in the brain. The relationship between brain copper levels and the neurological sequalae of WD is currently poorly understood.

In WD, reduced serum ceruloplasmin levels in patients result from reduced shuttling of copper from ATP7B to ceruloplasmin. Patients may also exhibit normal or only moderately reduced ceruloplasmin levels, for example, in the early clinical phase. However, changes in serum ceruloplasmin levels can be modified in both directions as results of inflammation and endocrine changes associated with pregnancy or hormonal-based contraceptives. Thus, the predictive value of serum ceruloplasmin levels for WD is debatable.

Although acute Coombs-negative acute hemolytic anemia may be observed in WD patients with acute liver failure and is included in the Leipzig scale, a false-negative Coombs test can occur in non-WD cases of acute liver failure. Thus, the value of this diagnostic sign is also unclear.

Undoubtedly, hepatic copper accumulation is a hallmark feature of WD, but other hepatic disorders, including cholestasis, can be associated with elevated copper levels. Further, copper is heterogeneously distributed in the liver, particularly in the setting of advanced fibrosis. As a consequence, there are potential sampling errors, and liver biopsies may not always be a reliable indicator of WD.

Measuring 24-hour urinary copper excretion is a valuable diagnostic method and is currently used for monitoring the efficacy of treatment. Symptomatic WD patients, especially those with neurological symptoms, exhibit markedly increased urinary copper excretion. However, children and asymptomatic WD patients may present with only slightly raised urinary copper levels. Patients treated with copper chelators initially exhibit markedly increased copper excretion, but upon treatment with zinc, a competitive copper uptake inhibitor, urinary copper excretion decreases.

Lastly, experimental models of WD have been developed to facilitate research on novel drug, genetic, and cellular therapies. Whereas these models typically develop liver disease, cerebral copper accumulation and associated neurological impairments are milder than those observed in patients, and thus, currently available models may not accurately represent the range of presentations of the human disease [75,76].

Recommendations for WD therapy were published in 2012 by the European Association for the Study of Liver (EASL) [74]. Given the challenges of diagnosis and the variable clinical presentation of this disorder, we remain far from the goal of achieving personalized medicine which addresses the needs of patients as individuals. The new guidelines will include the ongoing studies with new forms of trientine, molybdate, and more extended experience with zinc.

In summary, WD still poses many unresolved challenges regarding pathogenesis, diagnosis, neurological presentation, genetics, and copper. In this narrative a collection of eight case reports (Box 2, Box 3, Box 4, Box 5, Box 6, Box 7 and Box 8), plus one unusual patient (Box 1), we highlight these unresolved problems. In addition, Box 9 addresses the need to improve quantitative copper measurements, whereas the patients’ perspective is presented in Box 10. Lastly, the role of patient organizations is summarized in Box 11.

Box 4Neurological worsening on anti-copper treatment.Petr Dusek  A 27-year-old patient developed dysarthria, hand tremor, and cervical dystonia. He was diagnosed with Wilson disease based on positive Kayser-Fleischer rings, low serum ceruloplasmin, high urinary copper excretion, and bi-allelic *ATP7B* mutations [p.H1069Q]; [pH1069Q]. Liver biopsy documented cirrhosis, but his serum liver function tests were normal. Brain MRI demonstrated mild to moderate cerebellar and brainstem atrophy and a T2 hyperintense signal in the pons, mesencephalon, and thalami. No supratentorial atrophy was observed. His Baseline Unified Wilson’s Diseases Rating Scale (UWDRS) II/III sum score was 30. Chelation therapy with D-penicillamine was initiated five months after the onset of the first neurological symptoms. The dose was slowly increased to 450 mg daily within five months. During this period, the patient developed postural instability due to gait ataxia and leg dystonia while his hand ataxia and tremor progressed. His UWDRS score increased from 30 at baseline to 60 points in month 5. Due to deterioration of neurological functioning, the medication was adjusted to a combination of D-penicillamine (300 mg per day) and zinc sulphate (equivalent to 150 mg elemental zinc per day), but neurological symptoms further deteriorated. In month 11, his UWDRS score increased to 72 points, and treatment was changed to trientine monotherapy titrated to 1200 mg daily. Despite this, further neurological deterioration was observed; the patient developed severe dysarthria, dysphagia, weight loss, generalized dystonia, spastic paraparesis, vertical gaze palsy, and depression. At month 14, the patient became bedridden, and his UWDRS score reached 104 points despite urinary copper excretion rates of 762–19701 μg/24 h and serum free copper ranging from 0 to 37.5 μg/L. Treatment was adjusted to a combination of trientine (600 mg per day) and zinc (150 mg per day), and following intensive physical and speech therapy and nourishment via a feeding tub, e the patient gradually improved to the point that the feeding tube could be removed and the UWDRS score decreased to 56 points at month 20. Five years later, the patient condition has improved but is still partially dependent on caregivers; severe global brain atrophy and cavitations in the anterior thalamic nuclei were apparent on brain MRI, and the patient had an UWDRS score of 41 points.  This case, previously published in [77], illustrates that neurological deterioration on anti-copper treatment can occur despite a relatively short diagnostic delay and slow titration of copper chelators and may continue despite changes in treatment strategy. Neurological deterioration, despite timely introduction of currently recommended treatments, may lead to significant disability in 10–20% of WD patients [78].Outstanding questions:How can we identify WD patients who are at risk of neurological deterioration?Which strategies prevent, or attenuate the severity of, neurological deterioration after treatment initiation in WD patients?

Box 5Should we do brain MRI in WD patients without clinical neurological symptoms?Anna Czlonkowska  A 21-year-old man was admitted to our department to confirm the diagnosis of WD and to monitor anti-copper treatment. He was generally healthy. One year prior to diagnosis, he began working in a poorly ventilated hall where he was exposed to chemicals such as metals, solvents, and detergents. After 3 months, he noticed brown spots on both calves, and later, he noticed edema in his legs and experienced fatigue, which resulted in him stopping work after 10 months. Liver cirrhosis was diagnosed with esophageal varices, hypersplenism, and very low white blood cell count (WBC, ~3.0 × 10^9^/L) and platelets. WD was diagnosed based on low serum ceruloplasmin concentration and increased urinary copper excretion. A neurological examination was normal; however, brain magnetic resonance imaging (MRI) revealed T2 signal changes in the basal ganglia. He received 750 mg of D-penicillamine and 160 mg of zinc sulfate. Ten days later, he was transferred to our department to confirm the diagnosis and continue monitoring. Bilateral Kayser-Fleischer rings were detected. His neurological examination remained normal. However, soon after starting D-penicillamine therapy, the patient reported a mild slowness of speech and sialorrhea. The brain MRI confirmed T2 solid hyperintensity, indicating acute changes in caudate nucleus and putamen bilaterally. We decreased the dose of D-penicillamine to 250 mg daily and advised that the D-penicillamine dose should be slowly increased over 6 weeks to the final dose of 1000 mg daily. Zinc sulfate was withdrawn. He returned to us after a few weeks due to neurological symptoms—increased problems with speech and salivation, stiffness, and problems with walking. Neurological deterioration continued over the next few months. His face was masked with oromandibular dystonia, and he also developed hand and leg dystonia. He deteriorated markedly after a bout of pneumonia and needed assistance with nearly all daily activities. He could walk only with the help of two people, and a gastrostomy was implanted. The brain MRI after one year of treatment showed modest hyperintense and hypointense signals in the putamen with markedly hyperintense signals in lateral putaminal margins (typical for irreversible degenerative stage or putaminal necrosis). He has now been monitored for 5 years of treatment and is stable but is non-verbal with generalized dystonic symptoms and is wheelchair dependent. He uses a computer for communication and does not exhibit cognitive problems.Outstanding question:Should we do an MRI brain examination in patients with hepatic presentation, particularly in those with a normal neurological examination? We advocate this approach, because (i) MRI brain changes are detected in 40% of patients with the hepatic form of the disease [73], and (ii) the presence of changes in MRI is a predictor of early deterioration after starting treatment, especially with D-Penicillamine and lesions in thalami and pons [78].

Box 6Wilson disease must be carefully considered in the family of the index case, including parents.Anna Czlonkowska  A 31-year-old old female was referred to our clinic for confirmation of WD in her family. She suddenly developed severe hemolytic anemia and thrombocytopenia. Acute liver failure was diagnosed. She had a low ceruloplasmin level but neither Kayser-Fleischer rings nor neurological symptoms. She was treated with D-penicillamine, and her liver function improved while waiting for a liver transplant. Two pathogenic mutations (p.H1069Q/p.Q355X) were found. Her brother, age 36, was also investigated. He had a history of high alcohol consumption, a low platelet count, increased AST (151 U), AST (58 U), and exhibited hyperechogenicity of the liver. Ceruloplasmin was mildly decreased (0.13 g/L, normal range 0.15–0.30 g/L), and copper urinary excretion slightly increased (94 µg/24 h, normal range 8–80 µg/24 h). He was diagnosed as WD and received D-penicillamine. In our department, he was found to have a slightly decreased ceruloplasmin level (18.8 mg/ dL, norm 25–45 mg/dL) but only one mutation (pQ355X/−). We excluded WD following a radiolabeled Cu^64^ test [79]. One year after diagnosis of our proband, her father, at age 62, had pneumonia. During hospitalization for this, liver cirrhosis with esophageal varices was found but Kayser-Fleischer rings were absent. He received D-penicillamine. Following hospital admission, a moderate hand tremor was noted, and a subsequent MRI of the brain revealed discrete changes in the midbrain and pons, Kayser-Fleischer rings were now present, and ceruloplasmin levels were markedly decreased (3.0 mg/dL). Genetic analysis revealed the same mutations as his daughter. D-penicillamine was continued. He died one year later from cancer of unknown origin.Outstanding question:Should parents of WD index cases receive the same detailed screening (including DNA analysis) as siblings? We should carefully consider family history and include wider family members, not just siblings, in our investigations. As WD manifests mostly in young persons, detailed genetic studies of parents may not be required [80,81]. Nevertheless, carefully analyzing medical histories and basic laboratory tests are recommended. This is a rare case. WD was diagnosed in the parent of our index case; there is another case report of this type in the literature [82]. The pseudodominant (vertical) type of inheritance in WD is rarely diagnosed in our cohorts, being found in 4% of offspring [83].Additional problem for wider discussion:Siblings should undergo careful genetic testing, especially in the absence of significant changes in copper metabolism, even in the presence of liver disease, and must fulfill the Leipzig criteria [74]. Indications for genetic testing are given in [44].

Box 7Wilson Disease misdiagnoses: the role of liver copper quantification and genetic studies.Marina Berenguer and Carmen Espinós  In the absence of a single, reliable, and highly specific diagnostic test, diagnosis is based on the sum of clinical, laboratory tests, and genetic findings. Once diagnosed, treatment is based on either chelators (D-penicillamine and trientine) that enhance urinary copper excretion and/or zinc salts that decrease enteric copper absorption. If initiated before severe organ damage occurs and compliance is high, patient survival is similar to that of age-matched populations. Excellent outcomes are, however, not universal. Poor compliance with medication long-term is frequently associated with disease progression. In addition, incomplete resolution of symptoms and liver enzyme alterations can also occur, either related to lack of efficacy of available therapies and/or coexistence of comorbid conditions such as non-alcoholic steatohepatitis. We present a patient with presumed Wilson disease eventually found to be misdiagnosed, after years of WD therapy.  A man, aged 54, who had chronic elevation of liver enzymes for at least 5 years before referral to the specialist. The patient denied any type of alcohol consumption. No comorbid conditions or past family history of WD were reported by the patient. After extensive evaluations that excluded viral hepatitis (HCV, HBV), autoimmune liver diseases (Primary Sclerosing Cholangitis, Primary Biliary Cholangitis and autoimmune Hepatitis, hemochromatosis, and celiac or thyroid disease), the patient was suspected to have Wilson disease based on low ceruloplasmin levels (0.13 g/L) together with pathologically high liver copper levels (1766 µg/g DWL tissue) and slightly elevated urinary copper excretion (1-2 ULN) with a total Leipzig score of 4 (absence of Kayser-Fleischer rings, lack of neurological symptoms with normal brain MRI, no Coombs-negative hemolytic anemia). Genetic testing was not available at the time of diagnosis. The patient was started on zinc therapy with a slight improvement in liver enzymes, normalization of urinary copper excretion, and a calculated free serum copper lower than 10 ng/mL. As liver enzymes did not normalize despite adequate compliance, the patient was referred to our unit where treatment was changed to D-penicillamine one year later. Liver tests did not indicate any improvement, despite compliance assessed through urinary copper excretion and patient interviews. Due to the lack of liver response, together with relatively low copper excretion (always lower than 200 µg/24 h despite increased D-Penicillamine doses), a genetic evaluation was performed 3 years following the initial diagnosis. In this patient, we analyzed the *ATP7B* gene by Sanger sequencing and multiplex ligation-dependent probe amplification (MLPA) with no results of interest. We also performed exome sequencing without conclusive findings.  Given the negative findings in the genetic testing, together with the lack of changes in the liver, a decision was made to perform a new liver biopsy which revealed progression of liver fibrosis (from F1 to F3) together with steatohepatitis changes and a liver copper concentration of 18 µg/g DWL tissue. Of note, both measurements were performed in the same external specialized Wilson disease Unit. Following the second liver biopsy results, D-Penicillamine spell out was discontinued with no subsequent changes in laboratory parameters. A diagnosis of non-alcoholic steatohepatitis was made in the context of mild overweight (BMI 25.5).

Box 8Improving the quantification of copper overload in Wilson disease.Kay L. Double  There is current no available method to quantify tissue copper levels in living humans. This poses a major barrier to improving diagnosis and management of copper dyshomeostasis disorders, including Wilson disease. As exemplified in the case report outlined in Box 5, treatment of Wilson disease patients with chelation therapy, even by experienced neurologists, is challenging because our ability to monitor the efficacy of chelation therapy is poor. Whereas biofluids are used clinically as indirect indices of the effectiveness of chelation therapy, the validity of these indices in reflecting tissue copper levels is unclear. Using the best available data to date regarding brain and peripheral copper levels, we recently showed that there is no evidence that biofluid copper levels reflect those in the brain [84,85]. As the blood–brain barrier poses specific challenges for metal modification treatments, and neurological symptoms are of concern in Wilson disease and can worsen after the initiation of treatment, the ability to quantify and monitor brain copper levels longitudinally during life would be a major advance in our ability to safely and effectively treat this disorder [85].Outstanding questions:How can we better monitor the efficacy of copper chelation treatment during life?How are copper regulatory pathways altered in vulnerable organs, such as the liver and brain, in Wilson disease and do such changes explain variability in disease phenotype and in response to treatment?

Box 9A patient goes to the doctor.Louis C. Penning  A middle-aged “patient” entered the clinic with clear signs of hepatitis including jaundice, but no Kayser-Fleischer rings or neurological symptoms. Liver biopsy confirmed hepatic copper accumulation and cirrhosis. Family members were evaluated for clinical symptoms, liver changes, copper levels and genetic analysis. Genetic screens revealed the patient, and some close relatives, had a mutation in the *ATP7B* gene (p.R1453Q). Importantly, some relatives had an additional protective mutation in the *ATP7A* gene (p.T327I), which partially explained the clinical variation in hepatic copper levels in family members.  The “patient” described here is a Labrador retriever, a popular dog breed in the US and Europe which is predisposed for copper toxicosis due to mutations in the *ATP7B* gene. The high prevalence of copper toxicosis in this breed allowed us to perform canine-specific genome-wide association studies (GWAS) in cohorts of over 200 patients. This dog breed, or even dogs as a species, may be well positioned as a model to identify modifier genes related to hepatic copper levels because of the following advantages: large population numbers, availabilities of detailed pedigrees, state-of-art genetic tools and canine liver organoids for disease modeling. Similar mutations are found in Doberman pinchers. Notably, investigations of copper toxicosis in Bedlington terriers resulted in the discovery of the *COMMD1* gene.Outstanding questions:What modifier genes are related to variable hepatic copper levels and clinical presentation in WD patients? Can we use genetics of canine inbreeding to find modifier genes?What are the genetic predispositions to very rare copper-related disorders such as Indian childhood cirrhosis, endemic Tyrolean infantile cirrhosis and idiopathic copper toxicosis?

Box 10Often forgotten: the patient’s perspective.Wiebke Papenthin  A 22-year-old student feels depressed and finds she can no longer concentrate on her studies. She consults a general practitioner who prescribes psychotherapy, but this does not result in any improvement. On the contrary, her periods of depression become longer, and she is no longer able to maintain her studies. Following blood tests, the general practitioner transfers the patient to the local hospital where she is diagnosed with WD and chelation therapy is initiated. The patient’s condition deteriorates, and neurological symptoms develop associated with movement disorder, and the patient can no longer speak. Further neurological symptoms develop, and the dose of the chelator is varied but WD experts are not consulted. After a year of treatment, symptoms remain unchanged. The family contacted the patient organization who recommended consulting a neurological WD expert who diagnoses severe and irreversible neurological damage with a prognosis of life-long dependency on special care.  A 15-year-old teenager randomly diagnosed with WD at a young age who is clinically stable attends an appointment with his hepatologist, accompanied by his parents. A healthcare professional hepatologist discusses the well-being of the teenager with his parents. The mother reports that, while the teenager is compliant with his pharmacological treatment, he is less compliant with other aspects of treatment. For example, he eats copper-rich foods in secret in his room, which distresses his mother.  A middle-aged woman diagnosed with WD and several allergies reports to several of her healthcare professionals that she feels tired and is experiencing extremely low energy levels and an inability to concentrate. Her general practitioner suggests she may simply be lazy, while her hepatologist dismisses her concerns and tells her that the majority of her patients feel tired. Her allergist reports she is also often tired and contributes this to her age or stress. Her gynecologist suggests she could try hormone treatment if this does not improve.Outstanding questions:When can we trust the advice of our health professionals? When should we seek a second opinion? How can we help our health professionals reach an accurate and early diagnosis?What can we do to raise awareness of the need for expert centers? How can we ensure that healthcare professionals connect to such centers and ask for WD expert advice when required?How can we empower young patients to take responsibility for the management of their disease—and support parents of young WD patients as they learn these skills?How can we educate healthcare professional such that they take our concerns seriously and are open to seeking expert advice when needed?How can we teach healthcare professionals to listen to our concerns and see us as people, not just patients or a problem to be solved?

Box 11What can liver patients’ associations do for affected patients?Jose WillemseLiver patients’ associations have the aim to:Complement: The unique experience or ’hands-on’ expertise of patients complements the knowledge and experience of other parties in health care, such as healthcare providers, scientists and policymakers.Meeting the needs of patients: Patient participation ensures that developments in research, policy, guideline development, and care innovation are better attuned to patients’ needs and expectations. It also provides insight into what patients and their relatives consider feasible and acceptable.Contributes to implementation: Patient participation in all phases of a project or research also contributes to its implementation—that something is done with the findings.Furthermore, they:Follow the decision-making processes and ensure a democratic process of patient representation;Provide the perspective of patients on all relevant aspects in the policy & organisational processes;Promote and encourage, where possible, a patient-centric approach in both delivery of clinical care, service improvement and strategic development and decision-making;Advocate for care that is patient-centred and respectful of patients’ rights and choices;Provide the patient perspective on the application of personal data rules, compliance of information consent, and the management of complaints;Ensure that processes to address all ethical issues and concerns for patients are in place, balancing patient and clinical needs appropriately;Advise on transparency in quality of care, safety standards, clinical outcomes, and treatment options;Monitor the performance of the results by reviewing quality indicators such as the clinical outcomes of diagnosis and treatment;Contribute to the development and dissemination of patient information, policy, good practice, care pathways, and guidelines;Advise to a general project website with a lay version (in the native language of several partners);Contribute to research, e.g., defining research areas important to patients and their families and disseminating research-related information in a lay version by regular articles in their quarterly magazine concerning relevant information about the project, on websites, and other social media platforms such as Facebook and Twitter;Share the information with their colleague patient organizations related to the subject;Inform their umbrella organizations by articles/reports and presentations during international meetings;Advise about providing a lay version of the final results of the project.

## 6. Concluding Remarks

The Wilson Aarhus symposium, held 5–8 May 2022, was organized by Peter Ott (Aarhus, Denmark), Aurelia Poujois (Paris, France), Peter Ferenci (Vienna, Austria), Karl Heinz Weiss (Heidelberg, Germany), and Michael Schilsky (Yale-New Haven, USA). In that meeting, basic scientists and clinicians from many countries held an up-to-date, evidence-based, clinically orientated symposium with distinguished international faculty. Different expert groups, patient associations, and affected patients met and discussed major challenges in WD management and therapy. In particular, the benefits of new therapies for the treatment of WD were discussed, and the participants tried to establish a consensus on recommendations for future clinical WD trials. Based on these discussions and the examples presented in the previous boxes, a list of outstanding questions was agreed upon. These are summarized in Table 1.

## Figures and Tables

**Figure 1 biomedicines-11-00420-f001:**
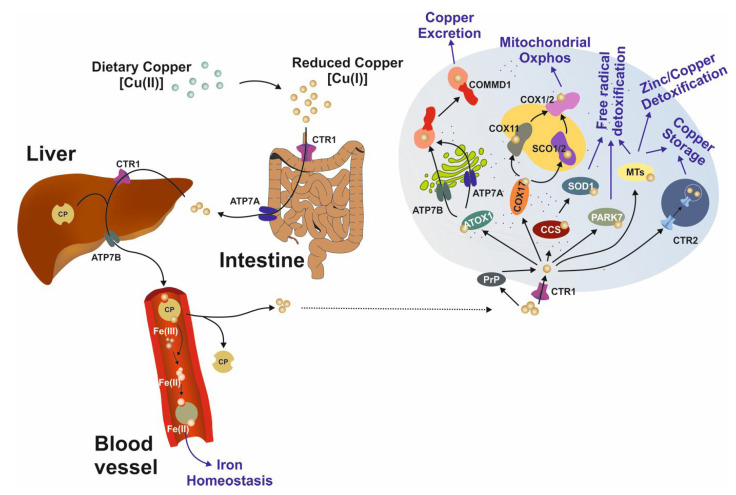
Schematic diagram of copper pathways and transport. Dietary copper is taken up in the gastrointestinal tract via CTR1 and excreted via ATP7A into the blood stream. It enters the liver via the portal vein and CTR1 mediates uptake into hepatocytes. Once bound to ceruloplasmin (CP), copper is excreted from the liver into the blood stream via ATP7B. The recycling of CP-released copper into the hepatocytes seems of very minor relevance (dotted line). A detailed image of intrahepatic copper transport is depicted on the right. Copper can be bound to ATOX1 and, following ATP7A- and ATP7B-mediated transport through the trans-Golgi network (TGN), copper binds to COMMD1 and is excreted. Intracellular copper in involved in mitochondrial oxidative phosphorylation by binding to COX17, COX11, SCO1/2, and COX1/2. Copper-initiated free radicals can be detoxified via the actions of CCS and SOD1, or alternatively by PARK7 and/or MT. Finally, intracellular copper can be stored in specific compartments via CTR2. Abbreviations: ATOX1, antioxidant protein 1; ATP7A, P-type ATPase 7A; ATP7B, P-type ATPase 7B; CCS, copper chaperone for SOD; COMMD1, copper metabolism murr1 domain 1;COX11, cytochrome C oxidase 11; COX17, cytochrome C oxidase copper chaperone 17; CP, ceruloplasmin; CTR1, copper transporter1; CTR2, copper transporter 2; MT, metallothionein(s); PARK7, Parkinson disease 7; PrP, Prion protein; SCO1/2, synthesis of cytochrome c oxidase 1/2; SOD1, superoxide dismutase 1. This figure is adapted from [16].

**Figure 2 biomedicines-11-00420-f002:**
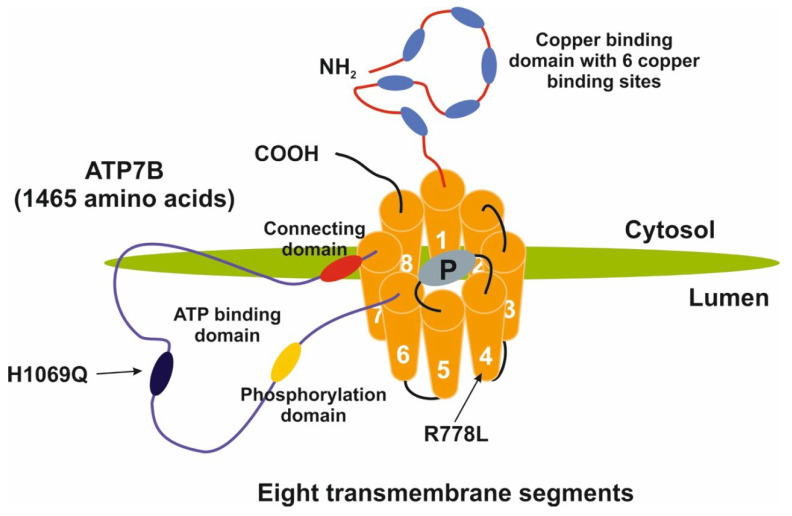
Schematic structure of the ATP7B copper transporter. The N-terminal domain of ATP7B contains 6 copper binding domains, and the eight transmembrane domains constitute a pore structure allowing copper to be transported through the membrane of the trans-Golgi network. The ATP-binding domain is located on the luminal side. The two most frequent mutations (p.H1069Q (US and Europe) and p.R778L (Asia) are indicated. A copper transport mechanism based on cryo-electron microscopy of *Xenopus tropicalis* ATP7B (highly homologous to human ATP7B) was recently published [26].

**Table 1 biomedicines-11-00420-t001:** The most important outstanding questions.

Topic	Question
Genetics	How can we best use DNA sequencing to improve WD diagnostics?
	Can we discover additional modifier genes?
Diagnosis	How can we improve monitoring the effects of copper chelation?
	How can we improve specific copper measurements?
	How can we improve diagnostics in neurological WD?
	Is the current gold standard for WD diagnosis sufficient to discriminate WD from other hepatic diseases?
	What is the role for autoimmune antibodies in WD?
Treatment	Should treatment and treatment monitoring be individualized based on patient age?
	What is the best approach for monitoring the efficacy of treatment for neurological WD?
Disease modeling	How are copper-regulatory pathways altered in WD?
	How relevant are organoid models?
Patient perspective	What is the best advice from healthcare professionals for patients?

## Data Availability

No new data were created or analyzed in this study. Data sharing is not applicable to this article.

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
