# Peer review of "A Century of Progress on Wilson Disease and the Enduring Challenges of Genetics, Diagnosis, and Treatment"

_biomedicines, 2023, doi:10.3390/biomedicines11020420_

Round 1

Reviewer 1 Report

This is an very interesting review on Wilson disease, written by Experts Physicians on this rare condition. The paper describes the state-of-the art regarding pathopysiology and diagnosis, whereas the sections about available treatments (pharmacological and not) are short and should be, in my opinion, widened (new drugs, management of hepatic and neurological complications, transplantation). I appreciate the clinical cases decribed by each Author, since they encourage discussion and highlight the diagnostic uncertainties we encounter in clinical practice.

There are few typos throughout the manuscript (especially in boxes) that should be revised.

I recommend the Authors to briefly describe what was the scope of the Meeting held in Aarhus and mentioned in the final part of the manuscript.

Finally, I congratulate the Authors because the patient's point-of-view has been highlighted. I think that a brief comment about not only patients, but also caregivers (especially for patients with neuropsychiatric involvment) should be added. 

Regards.

Author Response

Reviewer 1

This is a very interesting review on Wilson disease, written by Experts Physicians on this rare condition. The paper describes the state-of-the art regarding pathopysiology and diagnosis, whereas the sections about available treatments (pharmacological and not) are short and should be, in my opinion, widened (new drugs, management of hepatic and neurological complications, transplantation). I appreciate the clinical cases described by each Author, since they encourage discussion and highlight the diagnostic uncertainties we encounter in clinical practice.

Dear Reviewer 1,

Many thanks for taking the time to read our Concept paper and your overall encouraging words. As suggested we have removed several typos in the revised version of our article. All changes are marked in red letters. Please note that we have also reformatted the boxes.

There are few typos throughout the manuscript (especially in boxes) that should be revised.

I recommend the Authors to briefly describe what was the scope of the Meeting held in Aarhus and mentioned in the final part of the manuscript.

As suggested we have added some brief sentences about the scope of the Meeting held in Aarhus in the Conclusion section.

Finally, I congratulate the Authors because the patient's point-of-view has been highlighted. I think that a brief comment about not only patients, but also caregivers (especially for patients with neuropsychiatric involvement) should be added.

We fully agree that experts (basic scientists and clinicians) should closely work together with the patient’s organizations to identify the scientific and clinical needs that are associated with Wilson disease.

Reviewer 2 Report

This is a very well-written and interesting paper. It is educational and the findings are important and relevant. The writing is clear and concise. This paper is a very exhaustive review.

I have several observations and questions.

Line 243: ‘Elevation of liver enzymes…’ Should you add…’in the absence of concurrent liver pathology’?

Line 253: Should you briefly mention WD and pregnancy? Genetic counseling? Health risks for mother and fetus?

Author Response

Reviewer 2

This is a very well-written and interesting paper. It is educational and the findings are important and relevant. The writing is clear and concise. This paper is a very exhaustive review.

Dear Reviewer 2,

Many thanks for taking the time to read our Concept paper and your overall encouraging words. Please not that all changes in the revised version of our article are marked in red letters and that we have also reformatted the boxes to better follow the authors’ guidelines of the journals.

I have several observations and questions.

Line 243: ‘Elevation of liver enzymes…’ Should you add…’in the absence of concurrent liver pathology’?

As suggested we have added the additional information.

Line 253: Should you briefly mention WD and pregnancy? Genetic counseling? Health risks for mother and fetus?

Thanks for this comment. It is correct that pregnant women need close monitoring and multidisciplinary management because pregnancy induced complication in Wilson disease such as prematurity, hypertension, peeclampsia, and gestational diabetes mellitus are very frequent. Therefore, respective patients need adequate medical treatment and close monitoring before and during pregnancy to achieve best health outcome for mother and newborn. We have added a respective comment and cited an important paper reporting about complications during pregnancy of Wilson disease patients (new Ref.72). We have lined up all other references.